# Pelvic floor exercise: Awareness, knowledge, beliefs and practices among pregnant women in a Ghanaian setting

Yaa Abrafi Ankomah[1], Bright Anneh Awaitey[1]*, Moses Monday Omoniyi[1],
Benjamin Asamoah[1], Obed Kwame Numadzi[1], Kwofie Robert Amoah[1],
Joel Innocent Goli[2]

1 Department of Physiotherapy & Sports Science, Kwame Nkrumah University of Science and Technology,
Kumasi, Ghana, 2 St. John of God College of Health, Duayaw Nkwanta, Ghana

* brightanneh@gmail.com

## Abstract

### Background

Pregnancy and childbirth increase the risk of pelvic floor muscle damage, leading to conditions such as urinary incontinence, fecal incontinence, and pelvic organ prolapse. This makes it imperative to adopt strategies such as pelvic floor muscle exercises (PFMEs) to ensure positive antenatal and postnatal experiences for women. Hence, the study aimed to assess the awareness, knowledge, practices, and beliefs of PFMEs among pregnant women in a Ghanaian setting.

### Objective

We assessed the awareness, knowledge, practices and beliefs of PFMEs among pregnant women in a Ghanaian setting.

### Design

A cross-sectional descriptive research was conducted, recruiting 134 pregnant women through convenient sampling in selected antenatal clinics in Kumasi metropolis, Ashanti Region.

### Methods

The study adopted the questionnaire regarding pelvic floor muscle training among pregnant women used by Teerayut Temtanakitpaisan, Suvit Bunyavejchevin, Pranom Buppasiri and Chompilas Chongsomcha. Data on the socio-demographic characteristics of the women, their awareness, knowledge, beliefs and practices about PFMEs were recorded.

**Data availability statement:** All relevant data have been uploaded as Supporting Information files.

**Funding:** The author(s) received no specific funding for this work.

**Competing interests:** The authors have declared that no competing interests exist.

## Results

More than half (64.9%) of the respondents were not aware of PFMEs. Subsequent analysis for this work was therefore done for participants who reported being aware (35.1%). Also, the principal sources of information from which the subjects acquired the knowledge of PFME were health-care providers (42.6%) and media outside the hospital (36.2%). With reference to beliefs 80.3% believed that PFME could reduce vaginal trauma and 83.0% believe PFME aids vaginal birth. In terms of PFME practice, 36.2% of the aware women performed PFME regularly.

## Conclusion

To ensure pregnant women are well-informed about PFMEs, physiotherapists should work with antenatal care providers to develop comprehensive courses that include detailed information on PFME's.

## Introduction

Recent research has overwhelmingly shown that regular participation in PFME is generally safe and can contribute positively to both physical and mental well-being in pregnant women[1]. Nonetheless, the risk of pelvic floor muscle damage and weakness rises with pregnancy and childbirth in women [1]. For strengthening the pelvic floor muscles, the Kegel exercise and its variations are the most popular [2]. Among pregnant Ghanaian women, pelvic floor dysfunction is a prevalent problem, especially during the 3rd trimester and following childbirth [3]. Although there is evidence that pelvic floor muscle exercises (PFMEs) can prevent this problem, pregnant women in Ghana and other developing countries are not as likely to do PFMEs [4]. This has been attributed to numerous factors, such as a lack of awareness and information regarding PFMEs, socioeconomic hurdles to healthcare access, and inadequate training from healthcare providers [5]. Given the potential impact of PFME on maternal health and the limited research conducted in this area, there is a compelling need to assess the current awareness, understand the beliefs of pregnant women, and probe the level of practice towards PFME in a Ghanaian setting. This will provide a fundamental baseline for creating interventions to promote PFME practice.

### Pregnancy

Pregnancy is an intricate physiological process that involves the growth and development of a fetus inside the female uterus. The female body undergoes numerous physiological changes throughout pregnancy to sustain the growing fetus. Hormonal fluctuations, cardiovascular adaptations, metabolic changes, and immune system modifications are all part of these changes [6]. Hormonal variations, particularly higher levels of relaxin, contribute to ligament suppleness and pelvic joint instability, potentially causing pelvic floor dysfunction [7]. Furthermore, carrying a developing fetus can put mechanical strain on the abdomen, which can raise intra-abdominal

pressure and weaken pelvic floor muscles, making women more vulnerable to PFD including stress urine incontinence and pelvic organ prolapse [8].

## Pelvic floor muscle exercise

Pelvic floor dysfunction is the abnormal activity of the function of the pelvic musculature. This may result from impaired innervation of the pelvic muscle,obstetric or paturitional trauma, pelvic surgeries, advancing age or obesity. Fecal incontinence, urine incontinence, and pelvic organ prolapse all constitute Pelvic Organ Dysfunction [8]. PFME is suggested as the initial line of treatment to relieve pelvic symptoms in order to prevent and treat such damage [9]. PFME, also known as pelvic floor muscle training or pelvic floor rehabilitation, is a mainstream of behavioral treatment used for strengthening the pelvic floor muscles [10]. PFME is an exercise that targets the muscular strength of the pelvic floor including its power, endurance, relaxation, or a combination of these qualities [1]. The execution of Kegel exercises is not standardized. Training for workouts might range from brief verbal instructions to more intense sessions. Treatment options include printed materials, and individual sessions with a qualified therapist. The treatment intensity varies based on the number of training sessions and their frequency with which the patient practices exercise at home [11]. Research shows that after vaginal delivery, the incidence of pelvic floor dysfunction is related to PFM strength. Improving PFM strength can reduce the incidence of pelvic floor dysfunction within 6–12 months after delivery [11].

## Awareness, knowledge and practices related to PFME among pregnant women

Research shows that pregnant women's awareness of pelvic floor exercises varies. While some women express sufficient understanding of the benefits of pelvic floor exercises for aiding childbirth, minimizing urine incontinence, and fostering postpartum recovery, others show little understanding or have misconceptions about these exercises [12]. Healthcare providers play a significant role in expanding knowledge about pelvic floor exercises among pregnant women. During frequent prenatal visits, antenatal care providers, such as midwives, obstetricians, and physiotherapists, can educate women on the importance of pelvic floor exercises. Tantisiriwat and Manchana in 2014 discovered that most women who visited a gynecologic clinic had little knowledge, and half of the women were unaware of the effectiveness of PFME [13]. Though comprehension and knowledge of PFMEs are crucial for PFME performance, the results of a study in Nigeria show that pregnant women's knowledge of PFMEs does not guarantee that they will perform PFME practices [2].

PFMEs are widely recognized for their role in preventing and managing pelvic floor dysfunction. Despite awareness of the daily recommended frequency, only a minority of pregnant women engage in consistent PFME [14]. Pregnant women face several obstacles when it comes to performing pelvic floor exercises, even with its acknowledged advantages [15]. Time limits, lack of enthusiasm, and discomfort are frequently cited hurdles [16]. Physical discomfort, weariness, or conflicting demands on women's time and energy can all cause practice patterns to change throughout pregnancy [17].

Hence, this study aimed to assess the awareness, knowledge, practices and beliefs of PFMEs among pregnant women in a Ghanaian setting.

## Materials and methods

### Study design and participants

The study utilized a cross-sectional descriptive design in which a pretested structured questionnaire was administered to antenatal women. Convenient sampling was employed to recruit 134 women from Kumasi municipal, specifically from Kwame Nkrumah University of Science and Technology (KNUST) hospital and Emena Community hospital.

These facilities were purposively selected based on Ghana Health service classification of KNUST Hospital and Emena Community Hospital being respectively secondary and primary level of healthcare within the Kumasi metropolis. This thus allowed a diverse sample representation of pregnant women from different socioeconomic backgrounds.

An a priori power analysis was conducted using G*Power 3.1, based on a medium effect size (w = 0.30), an alpha level of 0.05, and a desired power of 0.80. A minimum sample size of 130 participants was required. The final sample of 134 participants exceeded this minimum, ensuring adequate power. Convenience sampling was employed during the study period, and participants who met the inclusion criteria and provided informed consent were enrolled.

## Inclusion and exclusion criteria

Pregnant women aged 18 years and above in their third trimester were eligible to participate in the study. No exclusions were made based on pelvic floor symptoms, urinary tract infections, pelvic organ prolapse, or other pelvic-related conditions.

Pregnant women who were not fluent in either English or Twi were excluded from the study. Additionally, pregnant women who declined to provide informed consent were excluded.

## Data collection instrument and research procedures

This study was conducted from 20th July 2024–30th September 2024. The questionnaire was adapted from [18] which was originally developed and validated among Thai pregnant women.

The questionnaire was reviewed and modified to ensure clarity and appropriateness for the Ghanaian context. Several items were simplified or reworded to improve comprehension. For example, the original question "If you know the benefits of pelvic floor exercise and the way to do the exercise, how often will you manage the pelvic floor muscles?" was split into two clearer questions: "Have you ever done pelvic floor exercise?" and "How often do you manage the pelvic floor muscles?" to better assess actual practice rather than hypothetical behavior.

A pilot study was conducted among 10 pregnant women who met the inclusion criteria but were not included in the main study. The pilot study assessed the clarity and comprehensibility of the questionnaire items, and participants provided feedback on their understanding of the questions. Based on this feedback, further minor modifications were made to improve question clarity. The Cronbach's alpha for the final questionnaire was 0.805, indicating good internal consistency.

The final questionnaire comprised four main sections: (1) socio-demographic characteristics (age, marital status, employment status, gestational age, parity); (2) awareness and knowledge of pelvic floor muscle exercises; (3) beliefs toward PFMEs; and (4) frequency of PFME practice. The questionnaire took approximately 7 minutes to complete and was administered in English. For participants who could not understand English, trained researchers orally translated the questions into Twi and recorded their responses.

## Statistical analysis

Statistical analysis was performed using the Statistical Package for Social Sciences (SPSS) software, version 21. Continuous variables (age, gestational age) were expressed as means with standard deviations (SD) and 95% confidence intervals (CI). Categorical variables (awareness, knowledge, beliefs, and practice of PFMEs) were summarized as frequencies, proportions, and 95% confidence intervals.

## Ethical considerations

Ethical approval (CHRPE/AP/619/24) was obtained from the Committee on Human Research, Publications and Ethics (CHRPE) at Kwame Nkrumah University of Science and Technology, Kumasi, Ghana, prior to participant recruitment and data collection.

All participants provided written informed consent before participating in the study. The consent process involved providing participants with detailed written and verbal explanations of the study's purpose, procedures, potential risks and benefits, confidentiality measures, and their rights as research participants. Participants were informed that their

participation was entirely voluntary and that they could withdraw from the study at any time without penalty. The signed written informed consent forms were retained by the research team in accordance with the CHRPE guidelines and the international research ethics standards.

## Results

### Participant's obstetric demographics

One hundred and thirty-four participants were recruited. Participants of the study had a mean age of 30.10±5.19 and 33.64±4.34 for gestational age. The mean gravidity, vaginal birth and caesarean operation were 2.26±1.42, 0.95±1.29 and 0.34±0.75, respectively. Table 1 below summarizes these participant's obstetric demographics.

### Awareness of PFMEs among participants

More than half (64.9%) of them were not aware of PFMEs. Subsequent analysis was therefore done for participants who responded to be aware (35.1%). The principal sources of information from which these subjects acquired the PFME information were health-care providers (42.6%) and media outside the hospital (36.2%). Table 2 below summarizes the number of participants aware of PFME in the studied population and the means via which PFME knowledge is acquired.

### Knowledge of PFMEs among participants

On the knowledge of PFMEs, 66% of them correctly identified the pelvic floor muscles as targeting around the genitals and 36.2% also correctly acknowledged normal breathing during these exercises. Only few 31.9% knew the abdominal muscle should not be contracted during PFMEs. Also, 38.3% recognized any position as acceptable for the exercises, and

**Table 1. Descriptive statistics of Participants' Continuous Demographic Variables (n = 134).**

|  | Mean | Standard deviation | 95% Confidence interval |
| --- | --- | --- | --- |
| Age | 30.10 | 5.192 | 29.22-30.98 |
| Gestational age | 33.64 | 4.335 | 32.91-34.37 |
| Gravida | 2.26 | 1.424 | 2.02-2.50 |
| Vaginal birth | 0.95 | 1.294 | 0.73-1.17 |
| Caesarean operation | 0.34 | 0.745 | 0.21-0.47 |

**Table 2. Awareness of PFMEs among the participants and means via which PFMEs knowledge is obtained.**

|  |  | Frequency | Percentage | Confidence interval (95%) |
| --- | --- | --- | --- | --- |
| Do you know about PFMEs? (n = 134) | Know | 47 | 35.1 | 27.0-43.2 |
|  | Don't know | 87 | 64.9 | 58.8-73.0 |
| Do you know about pelvic floor muscles? From who? (n = 47) | Husband | 1 | 2.1 | 0.1-11.3 |
|  | Family Member | 2 | 4.3 | 0.5-14.5 |
|  | Friend | 2 | 4.3 | 0.5-14.5 |
|  | Doctors | 2 | 4.3 | 0.5-14.5 |
|  | Hospital Personnel | 20 | 42.6 | 28.3-57.9 |
|  | Media in Hospital | 3 | 6.4 | 1.3-17.5 |
|  | Non-hospital media | 17 | 36.2 | 22.7-51.5 |

61.7% acknowledged the benefit of PFME as aiding easier birth. Notably, 87.2% correctly stated that pregnant women can exercise the pelvic floor (Table 3).

## Beliefs about PFMEs among participants

On the beliefs of PFME in relation to pregnancy outcomes (Table 4), 80.9% believed that PFME could facilitate delivery. In addition, 83.0% believed that PFME could reduce vaginal trauma during vaginal birth. However, 8.5% of the participants believed that PFME caused difficult vaginal childbirth or fetal demise. Regarding general health, 57.4% believed that PFME greatly improved their overall health, though very few participants believed that PFME could improve specific pelvic floor symptoms such as sexual dysfunction, pelvic pain, urinary incontinence or bowel dysfunction.

## Practices of PFMEs among participants

Regarding practice patterns (Table 5), the majority (46.8%) practiced PFME irregularly, while 36.2% practiced regularly and 17.0% had never practiced. Among those who practiced, 21.3% did so daily, while 46.8% practiced only sometimes. Easy birth was the primary motivation for exercising (68.1%). Factors helping consistency in PFME practice included easy birth (29.8%), seeing the benefits (25.5%), and other factors (29.8%).

**Table 3. Knowledge of PFMEs among the aware group participants (n = 47).**

| | | Frequency | Percentage | Confidence interval (95%) |
|---|---|---|---|---|
| Which area of muscles does the exercise target | Around genitals* | 31 | 66.0 | 50.9-79.0 |
| | Abdominal Muscles | 4 | 8.5 | 2.4-20.4 |
| | Back muscles | 1 | 2.1 | 0.1-11.3 |
| | Don't know | 11 | 23.4 | 12.3-38.0 |
| How should you breathe when exercising the pelvic floor muscles | Breath normally* | 17 | 36.2 | 22.7-51.5 |
| | Hold your breath | 5 | 10.6 | 3.5-23.1 |
| | Inhale and exhale | 9 | 19.1 | 9.1-33.3 |
| | Unknown | 16 | 34.0 | 20.9-49.3 |
| When exercising the pelvic floor muscle, should you contract the abdominal muscle | Should | 11 | 23.4 | 12.3-38.0 |
| | Should not* | 15 | 31.9 | 19.1-47.1 |
| | Don't know | 21 | 44.7 | 30.2-59.9 |
| Proper posture while exercising the pelvic floor muscles | Sit only | 16 | 34.0 | 20.9-49.3 |
| | Stand only | 1 | 2.1 | 0.1-11.3 |
| | Any position* | 18 | 38.3 | 24.5-53.6 |
| | Unknown | 12 | 25.5 | 13.9-40.3 |
| The benefit of exercising the pelvic floor muscle | Reduce incontinence | 5 | 10.6 | 3.5-23.1 |
| | Easy birth | 29 | 61.7 | 46.4-75.5 |
| | Not known | 13 | 27.7 | 15.6-42.6 |
| Can pregnant women exercise the pelvic floor muscles | Yes* | 41 | 87.2 | 74.3-95.2 |
| | No | 1 | 2.1 | 0.1-11.3 |
| | Unknown | 5 | 10.6 | 3.5-23.1 |

* indicates correct response.

**Table 4. Beliefs about PFMEs among Participants (n = 47).**

| Belief item | response | frequency | percentage | Confidence interval (95%) |
|---|---|---|---|---|
| **Pregnancy related beliefs** | | | | |
| Helps to give birth easier | Yes | 38 | 80.9 | 66.7-90.9 |
| | no | 4 | 8.5 | 2.4-20.4 |
| | Don't know | 5 | 10.6 | 3.5-23.1 |
| Help reduce vaginal tears | Yes | 39 | 83.0 | 69.2-92.4 |
| | No | 2 | 4.3 | 0.5-14.5 |
| | Don't know | 6 | 12.8 | 4.8-25.7 |
| Cause baby to die from lack of oxygen | Yes | 4 | 8.5 | 2.4-20.4 |
| | No | 36 | 76.6 | 62.0-87.7 |
| | Don't know | 7 | 14.9 | 6.2-28.3 |
| Cause preterm | Yes | 9 | 19.1 | 9.1-33.3 |
| | No | 30 | 63.8 | 48.5-77.3 |
| | Don't know | 8 | 17.0 | 7.6-30.8 |
| **Pelvic floor health beliefs** | | | | |
| General health | Greatly improves | 27 | 57.4 | 42.2-71.7 |
| | Slightly improves | 6 | 12.8 | 4.8-25.7 |
| | unchanged | 14 | 29.8 | 17.3-45.2 |

**Table 5. Descriptive Statistics of Practices of PFMEs among Participants (n = 47).**

| Practice Item | response | Frequency | Confidence interval (95%) |
|---|---|---|---|
| **Have you ever done pelvic floor exercise** | | | |
| Yes regularly | | 17 | 36.2 | 22.7-51.5 |
| Yes irregularly | | 22 | 46.8 | 32.1-62.0 |
| No | | 8 | 17.0 | 7.6-30.8 |
| **How often do you manage the pelvic floor muscles** | | | |
| Don't do it | | 13 | 27.7 | 15.6-42.6 |
| Sometimes | | 22 | 46.8 | 32.1-62 |
| 3-4 times a week | | 2 | 4.3 | .0.5-14.5 |
| Do it everyday | | 10 | 21.3 | 10.7-35.7 |
| **The motivation for exercising your pelvic floor muscles** | | | |
| No motivation | | 3 | 6.4 | 1.3-17.5 |
| Prevent incontinence | | 4 | 8.5 | 2.4-20.4 |
| Easy birth | | 32 | 68.1 | 52.9-80.9 |
| Others | | 8 | 17.0 | 7.6-30.8 |
| **Factors helping you manage the pelvic floor consistently** | | | |
| People remind regularly | | 4 | 8.5 | 2.4-20.4 |
| Seeing the benefits | | 12 | 25.5 | 13.9-40.3 |
| Do not want disease to be repeated | | 2 | 4.3 | 0.5-14.5 |
| Fear of illness | | 1 | 2.1 | 0.1-11.3 |
| Easy birth | | 14 | 29.8 | 17.3-45.2 |
| Others | | 14 | 29.8 | 17.3-45.2 |

## Discussion

In this study of pregnant women in Ghana, participants had a mean age of 30.1 years and an average gestational age of 33.6 weeks, with most having experienced at least one previous pregnancy. Despite this relatively mature and experienced obstetric profile, awareness and engagement with pelvic floor muscle exercises (PFMEs) were low. Only about one-third of participants could correctly identify body areas involved in PFMEs, and knowledge gaps persisted even among those who were aware, particularly regarding correct breathing patterns and posture. While many held positive beliefs about the benefits of PFMEs for delivery and pelvic floor protection, misconceptions about potential harm to the unborn child were also common. Regular practice was uncommon, with fewer than one-quarter reporting daily PFME performance. These findings highlight a critical gap in antenatal education and support for pelvic floor health, with implications for maternal well-being and postpartum recovery.

### Awareness of PFMEs among the participants

In this study, only 35.1% of respondents could correctly identify pelvic floor muscle exercises (PFMEs), indicating a substantial lack of awareness among pregnant women in Ghana. In Nigeria, a neighboring country, similar low awareness levels of 27.8% [19] and 46% [2] have been reported among pregnant women. These consistent findings across diverse LMIC settings suggest that low PFME awareness is a widespread issue in resource-limited contexts. However, comparable data on PFME awareness levels from high-income countries remains limited in the literature, making direct comparisons with high-income countries settings difficult. Nonetheless, these findings underscore the urgent need for enhanced education and awareness initiatives within antenatal care services in LMICs, ensuring that women receive timely and accurate information to support their pelvic floor health.

### Knowledge of PFMEs among the participants

Even among participants who were aware of PFMEs, knowledge was incomplete. The majority answered correctly on muscle type (66%), benefits (61.7%), and appropriate gestational timing (87.2%), yet substantial gaps remained regarding correct breathing patterns (63.8% incorrect) and posture during exercises (61.7% incorrect). This pattern of partial knowledge where a significant proportion of women possess general PFME knowledge alongside specific knowledge gaps has been consistently reported across both LMIC and HIC settings. For instance, studies in Nigeria [2] and Malaysia [20], (58%) have reported high proportions of participants with PFME knowledge. Similarly, in Western Australia, a HIC setting, Hill et al (2017) found that while 76% of pregnant women knew PFMs prevent urinary incontinence, there was incomplete knowledge regarding other functions such as faecal incontinence prevention (only 27%), and poor functional knowledge overall, with only 43% able to name more than one pelvic floor function [21]. These findings suggest that partial or incomplete PFME knowledge is a widespread challenge across diverse settings, rather than a problem unique to Ghana.

Such partial understanding raises concerns about the likelihood of correct PFME performance. Sigurdardottir et al (2020) reported that many women had knowledge of pelvic floor exercises but were unable to execute them effectively [22]. Woodley et al (2017) similarly emphasized that inadequate PFME knowledge represents a missed opportunity to improve women's health through physiotherapy [23].

In this study, healthcare providers were the most common source of PFME information (42.6%), followed closely by external media sources such as social media and other non-healthcare platforms (36.2%). The predominance of healthcare providers as information sources aligns with Hill et al. (2017) finding in Western Australia, where 49.4% of women received PFME information from midwives [21]. However, the notable proportion relying on informal sources in the current study (36.2%) suggests that PFME education may not yet be systematically integrated into routine antenatal care protocols in Ghana. This underscores the potential for well-designed, evidence-based social media interventions to complement provider-led education.

## Beliefs and perceptions among the participants

Most participants who were aware of PFMEs expressed positive beliefs about their benefits. Approximately 80.9% agreed that PFMEs could facilitate delivery, and 83.0% believed they could reduce vaginal trauma during labor, reflecting an appreciation of their role in protecting pelvic floor integrity. This pattern of positive beliefs is consistent with findings from other upper-middle-income settings, including Chen et al (2020) among Chinese women [24] and Alagirisamy et al (2022) among Malaysian women [25]. However, it contrasts with reports from Nigeria [19], a lower-middle-income country, where pregnant women held more negative beliefs about PFME benefits.

Despite the overall positive beliefs observed in this study, persistent misconceptions were also evident. Some participants believed that PFMEs might trigger premature birth or harm the unborn child. These safety-related misconceptions are not unique to Ghana. Ihegihu et al. (2022) found similar fears among pregnant women in Nigeria, who doubted the safety of PFMEs and worried about potential harm or induction of labor [26]. Additionally, Derrar et al. (2022) found that pregnant women in Saudi Arabia did not believe PFMEs had positive benefits to maternal health [27]. Such misconceptions and negative beliefs may act as barriers to PFME adoption, even when awareness exists. Addressing these concerns through clear, evidence-based counseling within antenatal care is essential to dispel myths and encourage safe and consistent practice.

## Practices of PFMEs among the participants

Only 21.3% of participants reported practicing PFMEs daily, indicating poor adherence despite the relatively high levels of positive beliefs documented earlier. This pattern of low PFME practice is common across diverse settings. Okeke et al. [2] and Temtanakitpaisanet al [18] found similarly poor PFME adherence among pregnant women in Nigeria and Thailand respectively. Even in high-income countries, adherence remains challenging. For example, Hill et al (2017) reported that only 11% of pregnant women in Australia practiced PFMEs regularly, despite higher baseline knowledge levels in that population [21]. The disconnect between positive beliefs and actual practice warrants closer examination.

Also, easy birth and physical proof of PFME benefits were common motivators for practice in this study. Some evidence suggests that higher PFME knowledge associates with increased practice [27] and that women with higher education and income demonstrate better PFME practice [28]. However, even when knowledge and motivation exist, barriers to adherence are well-documented across settings. Okeke et al (2020) found that Nigerian pregnant women cited tiredness, forgetfulness, and being too busy as primary reasons for non-adherence [2], which likely resonate with women in Ghana as well. Hay-Smith et (2016) similarly noted that despite understanding the benefits, many women struggled with motivation or lacked reminders to maintain regular practice [5].

The persistent gap between knowledge, positive beliefs, and actual practice across diverse settings including among educated populations in HICs, suggests that knowledge and beliefs alone are insufficient to drive sustained PFME practice. Supportive systems within routine antenatal care, such as onsite practical demonstrations of proper technique and take-home videos/charts, consistent reminders, and regular follow-up by healthcare providers during antenatal periods, may be necessary to help women translate their knowledge and positive beliefs into habitual exercise.

## Implications for practice

These findings highlight the need to include pelvic floor health education as a routine part of antenatal care in Ghana. Healthcare providers should be trained to teach PFMEs correctly, covering proper technique, breathing patterns, and posture. Educational materials should be in local languages and include anatomical diagrams and step-by-step guides to help women with varying levels of education understand better. Additionally, existing community health programs and antenatal group sessions could be used to encourage and normalize PFME practice among pregnant women, which could ultimately lead to better pelvic floor health outcomes for mothers.

## Limitations of the present study

This study acknowledges some limitations. Participants were recruited from only two hospitals in the Kumasi metropolis using convenience sampling, which may not fully represent all pregnant women in the region. Although the sample size met the minimum required, a larger sample from more facilities across the Kumasi metropolis would have provided more representative and generalizable findings. Additionally, the use of self-reported questionnaires may have introduced recall bias and social desirability bias, which could have led to overestimation of knowledge or practice levels. The findings may therefore have limited generalizability beyond the two study sites, and care should be taken when applying the results to other settings with different healthcare systems or cultural backgrounds.

## Conclusion

This study found low awareness, incomplete knowledge, persistent misconceptions, and poor adherence to PFMEs among pregnant women in Ghana. To address these gaps, physiotherapists should collaborate with antenatal care providers to develop structured education programs that include practical demonstrations, interactive sessions, and consistent follow-up during antenatal visits. Such programs should clearly explain the benefits, teach correct techniques, and dispel myths to empower women to integrate PFMEs into their daily routines. Strengthening PFME education in antenatal care has the potential to improve maternal pelvic floor health and postpartum recovery outcomes.

## Supporting information

**S1 File. Raw data.**
(XLSX)

## Acknowledgments

The authors express sincere gratitude to the pre-partum women who participated in this study and to the hospitals for approval of this study.

## Author contributions

**Conceptualization:** Yaa Abrafi Ankomah, bright Anneh Awaitey, Moses Monday Omoniyi, Benjamin Asamoah, Joel Innocent Goli.

**Formal analysis:** Obed Kwame Numadzi, Kwofie Robert Amoah.

**Investigation:** Yaa Abrafi Ankomah, Obed Kwame Numadzi, Kwofie Robert Amoah.

**Supervision:** bright Anneh Awaitey, Moses Monday Omoniyi, Benjamin Asamoah.

**Writing – original draft:** Yaa Abrafi Ankomah.

**Writing – review & editing:** bright Anneh Awaitey, Moses Monday Omoniyi, Benjamin Asamoah, Kwofie Robert Amoah, Joel Innocent Goli.

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
