## [Decision Letter · Decision Letter 0]

4 Jan 2026

Dear Dr. Awaitey,

Thank you for submitting your manuscript to PLOS ONE. After careful consideration, we feel that it has merit but does not fully meet PLOS ONE’s publication criteria as it currently stands. Therefore, we invite you to submit a revised version of the manuscript that addresses the points raised during the review process.

Thank you for submitting your manuscript. Both reviewers have provided detailed feedback and have suggested major revisions. Therefore, I invite you to respond to the reviewers' comments and submit your revised manuscript.

We look forward to receiving your revised manuscript.

Kind regards,

Sunita Panda, PhD

Academic Editor

PLOS One

Journal Requirements:

2. We note that your Data Availability Statement is currently as follows: “All relevant data are within the manuscript and its Supporting Information files.”

Reviewers' comments:

Reviewer's Responses to Questions

**Comments to the Author**

1. Is the manuscript technically sound, and do the data support the conclusions?

Reviewer #1: Yes

Reviewer #2: Partly

2. Has the statistical analysis been performed appropriately and rigorously?

Reviewer #1: Yes

Reviewer #2: No

3. Have the authors made all data underlying the findings in their manuscript fully available?

Reviewer #1: Yes

Reviewer #2: Yes

4. Is the manuscript presented in an intelligible fashion and written in standard English?

Reviewer #1: Yes

Reviewer #2: Yes

Reviewer #1: The authors have conducted a good study and the manuscript reflects a genuine effort. The topic selected is relevant and significant, and the methodology appears generally appropriate for the objectives outlined. I appreciate the clarity of writing, the logical flow of the information, and the attempt to interpret findings meaningfully in context of existing literature. Overall, this is a good piece of academic work, and the authors deserve acknowledgement for this contribution.

However, a few points need further clarification/explanation in order to strengthen the paper:

1.Clarification on Sample Selection

Point No.125

It would be helpful if the authors could elaborate on whether pregnant ladies with BOH, complaints of UTI, neurological deficit, or any pelvic related problems were excluded from the study. If yes, then it has to be specified in the exclusion criteria. This will help readers understand the generalizability of the findings.

2. Discussion on Results

Point No.138

The study findings are interesting, but a brief explanation of why only “Beliefs of 24 participants in section 3 towards PFME” were projected, when 47 subjects actually had knowledge about PFME. This explanation will enhance the clarity and avoid confusion in understanding the results. Rest tables are clear. Only this point is not clear.

3.Implication for Practice:

Could the authors expand a little further on how this study may be translated into clinical or public health practice? A short paragraph in the discussion would improve the practical value of the paper.

With these clarifications, the manuscript will be even stronger. I recommend revision addressing these points.

Overall, a good attempt and strong potential for publication after minor revisions.

Reviewer #2: Dear Authors,

The manuscript addresses an important topic in maternal health in Ghana, and the ethical conduct is clearly described. However, substantial revisions are required in the areas of methodology, statistical analysis, results presentation, and depth of discussion before the manuscript can be considered for publication.

Major revisions are required in following sections-

1. Study design and sampling- The use of convenience sampling from only two facilities, without a clear justification for site selection or a sample size calculation, substantially limits the generalizability of the findings and introduces potential selection bias. I recommend that the authors justify the choice of hospitals, present an a priori sample size estimation, and provide a sample recruitment flow (number approached, eligible, consenting, and included).

2. Questionnaire adaptation and validation- The manuscript states that a questionnaire was adapted from Temtanakitpaisan et al., but the adaptation and validation process for the Ghanaian context is not described in sufficient detail. The authors should explain how the tool was translated, culturally adapted, and pretested, and provide more information on the pilot study beyond reporting a Cronbach’s alpha (e.g., what was assessed, what changes were made).

3. Statistical analysis and reporting- The statistical methods are under described, and the analyses are almost descriptive. For transparency and rigor, the authors should

(a) specify all statistical tests used,

(b) report 95% confidence intervals for key estimates (means and proportions), and

(c) where possible, explore simple associations (e.g., between socio-demographic factors, awareness, and practice of PFMEs).

Clear denominators for all percentages in the tables are also needed.

4. Results consistency and clarity- There are inconsistencies in the reporting of sample sizes, particularly between the 47 women reported as “aware” of PFMEs and the 24 participants included in parts of the beliefs analysis, which are not explained. The authors should clarify how many women contributed data to each section and explain any missing data. I also suggest simplifying and restructuring the large tables (especially Tables 3–5) and considering a simple flow diagram to show how the final analytical samples were derived.

5. Depth of discussion and interpretation- The discussion section is relatively brief and does not fully engage with the broader literature or the implications of the findings. The authors are encouraged to more fully contextualize their results by comparing with similar studies in other low and middle income as well as high income settings, to explore in more depth why awareness and positive beliefs do not translate into practice, and to discuss the practical implications for antenatal care services and physiotherapy provision.

6. Limitations and reporting requirements- The limitations section is too narrow and should be expanded to include the implications of convenience sampling from two facilities, relatively small sample size, reliance on self-reported data (including possible social desirability bias), and limited generalizability beyond the study sites. In addition, the manuscript should include a data availability statement, funding information, a competing interests declaration, and an author contributions statement in line with journal policies.

Minor revision

1. Language and terminology- The manuscript would benefit from careful language editing to correct grammatical errors and improve clarity (e.g., “aides vaginal birth” should be “aids vaginal birth”; “mean of gravida” should be “mean gravidity”).

.

Reviewer #1: **Yes:**Prof. Delphina Mahesh GuravProf. Delphina Mahesh GuravProf. Delphina Mahesh GuravProf. Delphina Mahesh Gurav

Reviewer #2: No

---

## [Author Response · Author response to Decision Letter 1]

17 Feb 2026

COMMENTS RESPONSE Line location

EDITOR

Marked up copy of manuscripts with highlighted change- Uploaded

An unmarked version of revised paper without track changes - uploaded

Manuscript must meet PLOS ONE's style requirements, including those for file naming---Addressed using PLOS ONE style templates as guide

Data Availability Statement ---Uploaded as Supporting Information file

REVIEWER 1

Clarification on Sample Selection:

Elaborate on whether pregnant ladies with BOH, complaints of UTI, neurological deficit, or any pelvic related problems were excluded from the study.----Pregnant women were not excluded on pelvic floor disorders, UTI symptoms, or other pelvic-related problems. please refer to Line 136 -138 in manuscript

Discussion of results:

a brief explanation of why only “Beliefs of 24 participants in section 3 towards PFME” were projected, when 47 subjects actually had knowledge about PFME.---The stated 24 participants was an error. This has been deleted

=

Implication for Practice:

Expand a little further on how this study may be translated into clinical or public health practice. A short paragraph in the discussion would improve the practical value of the paper--This has been addressed as part of the discussions and a short paragraph has been introduced as well. please refer to Line 245 to 331 & 332 to 340 of manuscript

REVIEWER 2

Study design and sampling-

Justify the choice of hospitals, present an a priori sample size estimation, and provide a sample recruitment flow (number approached, eligible, consenting, and included)------Justification for site selection has been provided, incluing an a priori sample size calculation using G*Power 3.1 (minimum n=130; achieved n=134), and clarified our sampling approach in the revised methods section. please refer to Line 125 to 134

Questionnaire adaptation and validation-

The manuscript states that a questionnaire was adapted from Temtanakitpaisan et al., but the adaptation and validation process for the Ghanaian context is not described in sufficient detail. The authors should explain how the tool was translated, culturally adapted, and pretested, and provide more information on the pilot study beyond reporting a Cronbach’s alpha (e.g., what was assessed, what changes were made).---Detailed information on the questionnaire adaptation process has been provided now, including specific examples of how items were modified for the Ghanaian context (e.g., splitting complex questions into clearer ones). pilot study description has been expanded to explain what was assessed (clarity and comprehensibility), how feedback from pilot participants was handled. Refer to the revised methods section. Please refer to Line 143 to 165

Statistical analysis and reporting-

The statistical methods are under described, and the analyses are almost descriptive. For transparency and rigor, the authors should

(a) specify all statistical tests used,

(b) report 95% confidence intervals for key estimates (means and proportions), and

(c) where possible, explore simple associations (e.g., between socio-demographic factors, awareness, and practice of PFMEs).-------Clear denominators for all percentages in the tables are also needed. continuous variables were expressed as means with standard deviations and 95% confidence intervals, while categorical variables were reported as frequencies, proportions, and 95% confidence intervals. Clear denominators have been included in all tables. chi-square analyses was attempted to explore associations, but violated assumptions (expected frequencies <5) prevented reliable results thus was not reported in this study. please refer to Line 167 to 171, line 185 to 129

Results consistency and clarity- There are inconsistencies in the reporting of sample sizes, particularly between the 47 women reported as “aware” of PFMEs and the 24 participants included in parts of the beliefs analysis, which are not explained. The authors should clarify how many women contributed data to each section and explain any missing data. I also suggest simplifying and restructuring the large tables (especially Tables 3–5) and considering a simple flow diagram to show how the final analytical samples were derived.--------The 24 mentioned earlier was an error and this has been rectified. The participants. The number of women contributing to data for each section has been stated now. Tables 4 to 5 has been restructured and simplified. please refer to line 185 to 129; line 221 & 229

Depth of discussion and interpretation-

The authors are encouraged to more fully contextualize their results by comparing with similar studies in other low and middle income as well as high income settings, to explore in more depth why awareness and positive beliefs do not translate into practice, and to discuss the practical implications for antenatal care services and physiotherapy provision.-------- The discussion section has been expanded to include comparisons with studies from both LMICs and HICs. Gaps between awareness, beliefs, and practice have been explored in depth, and practical implications for integrating PFME education into routine antenatal care, including healthcare provider training amongst others, have also been discussed. please refer to Line 244 to 340

Limitations and reporting requirements-

The limitations section should be expanded to include the implications of convenience sampling from two facilities, relatively small sample size, reliance on self-reported data (including possible social desirability bias), and limited generalizability beyond the study sites. In addition, the manuscript should include a data availability statement, funding information, a competing interests declaration, and an author contributions statement in line with journal policies.-----The limitations section has been expanded to address convenience sampling from two facilities, sample size considerations, reliance on self-reported data with possible recall and social desirability bias, and limited generalizability beyond the study sites.

Reporting requirement such as Competing interests declaration, Data availability statement, funding information and author contributions statement have been addressed using the PLOS ONE style templates as guide

Language and terminology-

The manuscript would benefit from careful language editing to correct grammatical errors and improve clarity (e.g., “aides vaginal birth” should be “aids vaginal birth”; “mean of gravida” should be “mean gravidity”). This has been addressed. please refer to Line 37 and 187

---

## [Decision Letter · Decision Letter 1]

31 Mar 2026

Pelvic floor exercise: Awareness, knowledge, beliefs and practices among pregnant women in a Ghanaian setting

PONE-D-25-46176R1

Dear Dr. Awaitey,

We’re pleased to inform you that your manuscript has been judged scientifically suitable for publication and will be formally accepted for publication once it meets all outstanding technical requirements.

Kind regards,

Sunita Panda, PhD

Academic Editor

PLOS One

Additional Editor Comments (optional):

Reviewers' comments:

Reviewer's Responses to Questions

**Comments to the Author**

Reviewer #2: All comments have been addressed

Reviewer #3: All comments have been addressed

2. Is the manuscript technically sound, and do the data support the conclusions?

Reviewer #2: Yes

Reviewer #3: Yes

3. Has the statistical analysis been performed appropriately and rigorously?

Reviewer #2: Yes

Reviewer #3: Yes

4. Have the authors made all data underlying the findings in their manuscript fully available?

Reviewer #2: Yes

Reviewer #3: Yes

5. Is the manuscript presented in an intelligible fashion and written in standard English?

Reviewer #2: Yes

Reviewer #3: Yes

Reviewer #2: (No Response)

Reviewer #3: The authors have adequately addressed the concerns and observations raised in the previous round of review and I feel the manuscript can be accepted for publishing.

.

Reviewer #2: No

Reviewer #3: **Yes:**Prof. Delphina Mahesh GuravProf. Delphina Mahesh GuravProf. Delphina Mahesh GuravProf. Delphina Mahesh Gurav

---

## [Editor Report · Acceptance letter]

PONE-D-25-46176R1

PLOS One

Dear Dr. Awaitey,

I'm pleased to inform you that your manuscript has been deemed suitable for publication in PLOS One. Congratulations! Your manuscript is now being handed over to our production team.

Kind regards,

on behalf of

Dr Sunita Panda

Academic Editor

PLOS One